# Novel 2D and 3D Assays to Determine the Activity of Anti-Leishmanial Drugs

**DOI:** 10.3390/microorganisms8060831

**Published:** 2020-06-01

**Authors:** Alec O’Keeffe, Christine Hale, James A. Cotton, Vanessa Yardley, Kapish Gupta, Abhishek Ananthanarayanan, Sudaxshina Murdan, Simon L. Croft

**Affiliations:** 1Department of Infection Biology, London School of Hygiene and Tropical Medicine, London WC1E 7HT, UK; a.okeeffe@sygnaturediscovery.com (A.O.); vanessa.yardley@lshtm.ac.uk (V.Y.); 2Department of Pharmaceutics, UCL School of Pharmacy, University College London, London WC1N 1AX, UK; s.murdan@ucl.ac.uk; 3Sygnature Discovery, Bio City, Pennyfoot St, Nottingham NG1 1GR, UK; 4Wellcome Sanger Institute, Wellcome Genome Campus, Hinxton, Cambridge CB10 1SA, UK; cbh@sanger.ac.uk (C.H.); jc17@sanger.ac.uk (J.A.C.); 5Mechanobiology Institute, National University of Singapore, Singapore 117411, Singapore; Kapish.Gupta@Pennmedicine.upenn.edu; 6InvitroCue Pte Ltd., Singapore 138667, Singapore; ananth.abhishek@gmail.com

**Keywords:** *Leishmania*, macrophages, drug assays, media perfusion, 3D cellular models, iPSC macrophages

## Abstract

The discovery of novel anti-leishmanial compounds remains essential as current treatments have known limitations and there are insufficient novel compounds in development. We have investigated three complex and physiologically relevant in vitro assays, including: (i) a media perfusion based cell culture model, (ii) two 3D cell culture models, and (iii) iPSC derived macrophages in place of primary macrophages or cell lines, to determine whether they offer improved approaches to anti-leishmanial drug discovery and development. Using a *Leishmania major* amastigote-macrophage assay the activities of standard drugs were investigated to show the effect of changing parameters in these assays. We determined that drug activity was reduced by media perfusion (EC_50_ values for amphotericin B shifted from 54 (51–57) nM in the static system to 70 (61–75) nM under media perfusion; EC_50_ values for miltefosine shifted from 12 (11–15) µM in the static system to 30 (26–34) µM under media perfusion) (mean and 95% confidence intervals), with corresponding reduced drug accumulation by macrophages. In the 3D cell culture model there was a significant difference in the EC_50_ values of amphotericin B but not miltefosine (EC_50_ values for amphotericin B were 34.9 (31.4–38.6) nM in the 2D and 52.3 (46.6–58.7) nM in 3D; EC_50_ values for miltefosine were 5.0 (4.9–5.2) µM in 2D and 5.9 (5.5–6.2) µM in 3D (mean and 95% confidence intervals). Finally, in experiments using iPSC derived macrophages infected with *Leishmania*, reported here for the first time, we observed a higher level of intracellular infection in iPSC derived macrophages compared to the other macrophage types for four different species of *Leishmania* studied. For *L. major* with an initial infection ratio of 0.5 parasites per host cell the percentage infection level of the macrophages after 72 h was 11.3% ± 1.5%, 46.0% ± 1.4%, 66.4% ± 3.5% and 75.1% ± 2.4% (average ± SD) for the four cells types, THP1 a human monocytic cell line, mouse bone marrow macrophages (MBMMs), human bone marrow macrophages (HBMMs) and iPSC derived macrophages respectively. Despite the higher infection levels, drug activity in iPSC derived macrophages was similar to that in other macrophage types, for example, amphotericin B EC_50_ values were 35.9 (33.4–38.5), 33.5 (31.5–36.5), 33.6 (30.5—not calculated (NC)) and 46.4 (45.8–47.2) nM in iPSC, MBMMs, HBMMs and THP1 cells respectively (mean and 95% confidence intervals). We conclude that increasing the complexity of cellular assays does impact upon anti-leishmanial drug activities but not sufficiently to replace the current model used in HTS/HCS assays in drug discovery programmes. The impact of media perfusion on drug activities and the use of iPSC macrophages do, however, deserve further investigation.

## 1. Introduction

Leishmaniasis is a neglected tropical disease caused by protozoan parasites of 20 different species of *Leishmania*. The parasite has two distinct life cycle stages, the motile promastigote, which is found extracellularly in the gut of the sand fly vector, and the amastigote, an intracellular stage that survives and divides in the phagolysosomal compartment of macrophages. There are two main disease manifestations of leishmaniasis, the potentially fatal visceral form (VL) and the self-curing but disfiguring cutaneous form (CL), which together have an incidence of 1 to 1.5 million cases in over 100 endemic countries [1,2,3,4,5]. The current drugs for the treatment of both forms of leishmaniasis have well defined limitations [6,7] and although there are several novel compounds in development [8,9] (https://www.dndi.org) pipeline, given the known high attrition rates in drug development, there remains a need for further novel compounds to be added to the pipeline. Since the early 1980s the in vitro amastigote macrophage model has been the standard assay for drug discovery [10,11]; this model has provided the basis for high throughput and high content screening assays to identify novel anti-leishmanial compounds [12,13]. These assays are all based upon a macrophage monolayer system with exposure to compounds in culture medium for defined periods at a defined temperature. This basic assay model for anti-leishmanial discovery has not changed over the past decades in contrast to assays for several other infectious diseases, for example malaria [14], amoebiasis [15] and tuberculosis [16] where relevant physiological factors or more appropriate host cells have been used. Our aim was therefore to investigate whether changes in assay design could expedite the drug discovery process or add to lead identification or lead optimization stages. Specifically, we determined the influence of: (a) media flow, (b) 3D cell culture models and (c) use of differentiated human induced pluripotent stem cells (iPSC) as the host cells on the activity of standard anti-leishmanials.

The biological role that fluid flow plays in physiology has been recognized as significant for many decades now, whether it is blood flow, media flow or interstitial flow [17]. The perfusion of culture media in a cell culture system allows for increased nourishment and sustainability of cultures, enables blood and lymphatic capillary morphogenesis in vitro [18,19,20], the functional activity of chondrocytes and osteocytes [21,22], fibroblast differentiation [23] and induction of cytokine production by smooth muscle cells [24]. Static systems do not offer any form of dynamic chemical or physical stimuli to cells, such as gradients in chemical concentrations, flow, pressure, or mechanical stress caused by movement of fluids around them. This is a major limitation in experiments investigating cellular responses in vitro as the complex interplay of mechanical and biochemical factors is absent [25]. Fluid flow is also important for drug delivery to cells and can be used to model the permeation of drugs to a tissue, including chemical gradients or the pulsatile nature of drug delivery [26] and the clearance of potentially toxic metabolites [27].

2D cell culture systems have been used for in vitro assays to measure infection rates and drug efficacies against *Leishmania* parasites since 1975 [28]. In 2D cell culture systems, cells are plated out into wells and left to settle and attach to a flat surface, spreading and becoming wider and flatter with a rearrangement of their internal structure to reflect this environment. In contrast, cells found within tissues have a 3D conformation or assemble into a 3D architecture that can more accurately reproduce the anatomy or physiology of a tissue for more informative studies [29,30] with, not just an altered surface area to volume ratio but also changes in cell-cell interactions [31] and receptor presentation [31] which could affect pathogen invasion [32] and drug/particle/molecule uptake [33].

Over the past decades a wide range of cell types have been used in assays to determine anti-leishmanial drug activity, from the Sticker’s sarcoma dog fibroblast cell line [28], primary isolated murine peritoneal macrophages (PEMs) and human monocytes, transformed rodent macrophage cell lines [34] and a human monocytic cell line, derived from an acute monocytic leukaemia patient (THP1) [35], the latter having the advantage of potentially unlimited division and easy maintenance in high content (HCS) and high throughput screens (HTS). Cell lines have cancer-like properties, such as immortalisation but also other phenotypes [36]. A disadvantage is that these cells carry mutations that are responsible for their immortality, making them different from the primary cell that they represent [37]. Primary isolated cells such as PEMs or mouse bone marrow macrophages and human monocyte-derived macrophages have been used extensively but only maintain their functional similarity to macrophage cells within the body for a short period [38] and are not suitable for provision of the large number of cells required for screening compound libraries. In 2006, Takahashi and Yamanaka [39] showed that the forced expression of four transcription factors (Oct4, Sox2, Klf4, and c-Myc) was sufficient to convert fibroblast cells into embryonic stem cell-like cells, induced pluripotent stem cells (iPSC). Since then a variety of starting cell types, different combinations of main transcription factors and techniques to deliver the transcription factors into cells have been used successfully. Many cell types have been derived from iPSCs such as macrophages and cardiomyocytes but their real utility in drug discovery has not been elucidated.

Here we show how adding complexity to the *Leishmania* amastigote–macrophage assays alter responses to standard anti-leishmanial drugs (Table 1) and consider whether this complexity improves the likely predictivity of the assays. The overall goal would be for these additional layers of complexity to be integrated into the standard in vitro assay formats to better predict in vivo data.

## 2. Materials and Methods

All materials sourced from Sigma (Sigma Aldrich, Gillingham, Dorset, UK) unless otherwise stated.

### 2.1. Drugs 

The standard drugs used in this study are listed in Table 1.

### 2.2. Leishmania Parasites

*Leishmania major* (MHOM/SA/85/JISH118) and *Leishmania mexicana* (MNYC/BZ/62/M379) amastigotes were obtained and isolated from BALB/c mouse skin lesions. They were allowed to transform to promasigotes and were maintained in Schneider’s insect medium supplemented with 10% heat inactivated foetal calf serum (HiFCS) (Harlan, Bicester, Oxfordshire, UK) at 26 °C. *Leishmania amazonensis* DSRed (IFLA/BR/1967/PH8) (obtained from Dr Eric Prina, Institut Pasteur, Paris, France) and *Leishmania major* mCherry (LV39c5/RHO/SU/59/P) (obtained from Prof Rosa Reguera, University of Léon, Leon, Spain) were maintained in M199 medium supplemented with 10% HiFCS at 26 °C.

The parasites were routinely passaged through BALB/c mice (Charles River, Saffron Walden, UK) and low passage number promastigotes (below passage number 3) were used for these experiments due to the loss of infectivity with time of parasite cultivation [40].

All animal work was carried out under a UK Home Office project licence according to the Animal (Scientific Procedures) Act 1986 and the new European Directive 2012. The Project Licence was reviewed by an Animal Welfare and Ethical Review Board prior to submission and consequent approval, to the UK Home Office. 

### 2.3. Macrophage Types

#### 2.3.1. THP1 Cells

THP1 cells (ATCC, via LGC standards, Teddington, Middlesex, UK) (TIB-202™) were maintained in RPMI-1640 medium supplemented with l-glutamine and 10% HiFCS. The THP1 cell line was maintained in an incubator at 37 °C and 5% CO_2_ and passaged to new medium once a week (1/10 *v*/*v* ratio of cells to fresh medium). THP1 cells were differentiated for 24 h at 37 °C 5% CO_2_ from monocyte to macrophage by the addition of 20 ng/mL Phorbol 12-myristate 13-acetate (PMA).

#### 2.3.2. Mouse Bone Marrow Monocytes 

Bone marrow-derived macrophages (BMM) were obtained from femurs of female BALB/c mice (Charles River Ltd., Saffron Walden, Essex, UK). A 25 g needle was used to flush the bone cavity with 5 mL of ice-cold Dulbecco’s modified Eagles medium (DMEM). Flushed DMEM was collected and then centrifuged before re-suspending the bone marrow progenitor cells in RPMI 1640 + 10% HiFCS and penicillin/streptomycin (pen/strep).

The mouse bone marrow monocytes were plated at a concentration of 25 million cells per T175 flask. Then extra RPMI + 10% HiFCS and penicillin/streptomycin containing macrophage colony stimulating factor (m-CSF) was added to give a final concentration of 50 ng/mL of m-CSF. The cells were incubated for 7 days in a 37 °C incubator before mature macrophages were harvested.

#### 2.3.3. Human Bone Marrow Monocytes

Human bone marrow mononuclear cells from a single donor were purchased from the American Type Culture Collection (ATCC, VIA LGC standards, Teddington, Middlesex, UK) (PCS-800_013). 25 million cells were delivered cryopreserved, these were defrosted and plated. All experiments were conducted on the same batch of cells. The ATCC have characterised these cells to be CD45, CD3, CD8, CD58, CD14, CD19 and CD34 positive immediately before freezing.

All of the human bone marrow monocytes were plated in a T175 flask in RPMI 1640 + 10% HiFCS, pen/strep and 50 ng/mL m-CSF. The cells were incubated for 7 days in a 37 °C incubator before mature macrophages were harvested.

#### 2.3.4. iPSC Cells 

The human iPSC lines used in this study, Kolf2, were generated by the Sanger Institute’s Human Induced Pluripotent Stem Cells Initiative (HipSci) project and can be purchased via Public Health England/ECACC (HPSI0114i-kolf_2, catalogue number 77650100). To differentiate human iPSCs into macrophages, we used the approach of Hale et al. [41]. Briefly, a subclone of undifferentiated human iPSCs Kolf2 C1 were grown from frozen vials, initially feeder-free on gelatin coated plates in TESR-E8 medium and then on mitotically inactivated mouse embryonic fibroblast (MEF) monolayers. Once grown to confluence the cells were collected and transferred into iPSC base medium (Advanced DMEM/F12 supplemented with 20% KnockOut Serum Replacement, 2 mM L-Glutamine, 0.055 mM β-mercaptoethanol) in 10cm^2^ bacterial petri dishes for 4 days to generate Embryoid Bodies (EBs). On Day 5, the ball-like EBs were transferred back into 10cm^2^ tissue culture dishes that had been pre-coated with gelatin. Myeloid precursor cells were generated in the presence of 25 ng/mL IL-3 (R&D, now Bio-techne, Abingdon, Oxfordshire, UK) and 50 ng/mL m-CSF (R&D) followed by terminal differentiation and maturation of myeloid precursors into matured macrophages in the presence of m-CSF (1000  ng/mL).

#### 2.3.5. Mouse Peritoneal Macrophages (PEMs)

Mouse peritoneal macrophages were isolated from female CD1 mice 24 h after peritoneal injection with 0.5 mL of 2% starch solution in sterile water. Macrophages were collected by abdominal lavage with cold RPMI-1640 medium containing 1% penicillin and streptomycin. The collected cells were centrifuged for 15 min at 500 g and 4 °C, washed in RPMI-1640 medium and re-suspended in RPMI-1640 medium containing 10% HiFCS.

### 2.4. Effect of Perfusion of Culture Medium on Anti-Leishmanial Activity of Drugs

The influence of perfusion of the culture medium on the infection levels of *Leishmania* was investigated using the QV900 media perfusion system (Kirkstall LTD, York, UK) developed and detailed in elsewhere [42]. The method for the static infection of PEMs seeded on 12 mm glass cover slips and their transfer to the QV900 is also described in O’Keeffe et al. [42]. Briefly, PEMS are seeded onto glass coverslips and infected with promastigotes in static conditions for 24 h. Subsequently two thirds of the glass coverslips containing infected macrophages were transferred to the media perfusion system and maintained under flow conditions either at the bottom of the well or on top of an insert, at a flow speed of 360 μL/min for 72 h in a 34 °C, 5% CO_2_ incubator. The remaining coverslips were used for the static control, with macrophages maintained in the same culture medium without flow. This creates three different culture media flow rates, (i) static (0 m/s), (ii) low flow (1.45 × 10^−9^ m/s; cells placed at bottom of well) and (iii) high flow (1.23 × 10^−7^ m/s; cells placed on top of inserts) [42]. After the initial 24 h of static infection, the media used for the activity of drug studies was supplemented with drug (miltefosine at 20 μM, 5 μM or 1.25 μM, amphotericin B (AmB) at 200 nM, 50 nM or 12.5 nM, sodium stibogluconate at 600 μg, 200 μg or 60 μg of Sb^V^/mL or paromomycin sulphate salt at 300 μM, 100 μM or 30 μM). After 72 h, all coverslips were removed from the wells, methanol-fixed and Giemsa stained, and microscopically examined (400× magnification) to count the number of infected macrophages and thereby measure the drug-induced reduction in parasite infection of the host cell. Non-linear sigmoidal curve fitting (variable slope) was conducted using Prism Software (GraphPad, Surrey, UK). 

### 2.5. Effect of Perfusion of Culture Medium on Drug Accumulation by Infected Macrophages

Experiments were conducted as described in the previous section, except for the drug concentrations and time points used. In these experiments, miltefosine at 20 μM or AmB at 1 µM were added to each well. Cultures were maintained for 4, 8, 12 or 24 h at 34 °C and 5% CO_2_, after which coverslips were transferred to a new 24 well plate and washed three times with cold PBS. Drug extraction methods were followed from Voak et al. [43]. When miltefosine was used, a back-exchange step was conducted by adding 500 µL of 3% (*w*/*v*) fatty acid-free BSA in PBS to remove any membrane-bound miltefosine, and then washing the cells again with PBS. Subsequently, the cells were lysed by the addition of 0.1% (*v*/*v*) formic acid in water to each well, and vigorous mixing by pipetting up and down every five minutes during a 30 min incubation at room temperature.

Lysates were transferred to a microcentrifuge tube with 250 µL acetonitrile with tolbutamide (200 ng/mL). When amphotericin B was used, cells were lysed (as described for miltefosine), and lysates were transferred to a microcentrifuge tube with 250 µL of 16% DMSO in methanol with tolbutamide (200 ng/mL). Tolbutamide was used as an internal standard for both drugs. Microcentrifuge tubes containing the internal standard and miltefosine or amphotericin B were placed on a shaker for 10 min before being centrifuged for 15 min at 4150 g at 4 °C. 200 µL of the supernatant was transferred into 96 well plates and stored at −80 °C. A calibration curve was created using untreated cell lysates spiked with known miltefosine or amphotericin B concentrations. A further set of blanks were prepared without tolbutamide [43].

Drug concentrations were determined by Pharmidex Ltd., a company specialising in ADMET assays with experience in miltefosine and amphotericin B drug accumulation in cell studies [43] using HPLC-MS-MS with electro-spray ionisation on an Agilent 1200 HPLC/Agilent 6410 triple quad under positive ion MS-MS mode. Data were extracted into Excel files and analysed in both Excel and Prism. The remaining lysate was used to measure protein concentration as a measure of cell number using a Pierce™ BCA protein assay kit (Thermo Scientific, Loughborough, UK).

### 2.6. 3D Cell Culture Experiments

#### 2.6.1. 3D Cell Seeding Protocols

Alvetex scaffolds (24 well sized spun polystyrene scaffolds (ReproCELL Europe Ltd., Stockton-on-Tees, UK) were submerged in 70% ethanol for 10 s, and then washed in RPMI-1640. Both Invitrocue (3D CelluSponge, Invitrocue Ltd., Singapore 138667, Singapore) and Alvetex Scaffolds were then placed in a 24 well plate. PEMs were washed, centrifuged and re-suspended at 4 × 10^7^ per mL in RPMI-1640 supplemented with 10% HiFCS. Cells were seeded into Invitrocue or Alvetex scaffolds by pipetting 25 µL of cell suspension directly on to the scaffold (as specified by Invitrocue for the CelluSponge). Scaffolds were maintained at 37 °C and 5% CO_2_ for 45 min, after which the wells were topped up with 500 µL of RPMI-1640 supplemented with 10% HiFCS and replaced in the incubator for 24 h. 

#### 2.6.2. Infection by *Leishmania* in 3D Cultures

The adhered PEMs in the 3D scaffolds were infected with stationary phase promastigotes, at a range of parasite: peritoneal macrophage ratios (0.5:1 to 6:1). The scaffolds were maintained at 34 °C and 5% CO_2_ for a further 72 h, after which they were fixed in 4% paraformaldehyde (PFA) overnight at 4 °C. After 24 h, the scaffolds were treated with 0.2% Triton X-100 in PBS for 10 min and then 1% BSA in PBS for 10 min. Scaffolds were then stained with phalloidin Cruz fluor Actin labelling antibody (Santa Cruz Biotechnology, Heidelberg, Germany) at 4 °C on a plate shaker for 24 h, after which they were washed and treated with 300 mM DAPI stain for 10 min. The scaffolds were examined using a Zeiss LSM510 confocal microscope (40× magnification) (Zeiss, Germany) and the number of macrophages that were infected (out of a total of 100 macrophages) were counted to determine the percentage infection. Images taken during the course of the experiments were analysed computationally both at Invitrocue (using the ImarisCell module of Imaris 8.2.0 (Oxford Instruments-Bitplane AG, Zurich, Switzerland)) and at LSHTM (using the Volocity software, Quorum Technologies, https://quorumtechnologies.com/), to produce values for percentage infection and the number of parasites per cell.

#### 2.6.3. Drug Activity in 3D Infection Models

The adhered PEMs in the Invitrocue scaffold were infected with stationary phase promastigotes at a ratio of 5 *L. major* (mCherry) promastigotes or 7 *L. amazonensis* (DSred2) to 1 macrophage and maintained at 34 °C in a 5% CO_2_ for 24 h. After 24 h, media was removed from all wells and miltefosine at concentrations of 20 μM, 5 μM and 1.25 μM or amphotericin B at concentrations of 200 nM, 50 nM and 12.5 nM were added in triplicates. A set of drug-free controls were included in triplicate, by adding culture media. After 72 h, percentage infection of PEMs was assessed as described above. This percentage infection was then scaled in comparison to the untreated and uninfected controls.

### 2.7. Influence of Host Cell Type on Infection by Parasite

To investigate the potential of iPSC cells we first compared infection levels in macrophages of different origin, namely THP1, human and MBMM and iPSC derived macrophages within standard static 2D assays. The cells were seeded in 16-well Lab-Tek™ plates (Thermofisher, Paisley, UK) in RPMI-1640 supplemented with 10% HiFCS at a density of 2 × 10^4^ per well. After 24 h incubation at 37 °C and 5% CO_2_, cells were infected with stationary phase promastigotes *(L. major* JISH 118, *L. major* mCherry, *L. mexicana* M379 and *L. amazonensis* DSred2) at a range of parasite: host cell ratios (0.5:1 to 10:1) and cells were maintained at 34 °C in a 5% CO_2_. After 24 h, the culture medium was replaced to remove extracellular promastigotes and one slide was fixed with methanol and stained with Giemsa to determine the initial level of infection. Infection of host cells was determined microscopically (400× magnification) by counting the number of macrophages that were infected and the number of amastigotes per host cell (out of a total of 100 macrophages).

In cultures with a sufficient level of infection, (i.e., >50% of macrophages being infected after 24 h) a drug (miltefosine at 20 μM, 5 μM or 1.25 μM, amphotericin B at 200 nM, 50 nM or 12.5 nM, sodium stibogluconate at 600 μg, 200 μg or 60 μg of Sb^V^/mL, paromomycin sulphate salt at 300 μM, 100 μM or 30 μM) was added (in triplicate on the same plate) and incubation was continued for a further 72 h. All slides were methanol-fixed, Giemsa stained, and microscopically examined (400× magnification) to count the number of macrophages that were infected (out of a total of 100 macrophages) and thereby measure the drug-induced reduction in parasite infection of the PEMs. The average number of amastigotes per peritoneal macrophage were also counted.

### 2.8. Statistical Analysis

All statistical analysis was performed using Graphical Prism Vs7 (Graphpad, San Diego, CA, USA).

## 3. Results

### 3.1. The Influence of Culture Media Perfusion on Anti-Leishmanial Drug Activity

The activities of amphotericin B (Figure 1a), miltefosine (Figure 1b), paromomycin sulphate (Figure 1c) and sodium stibogluconate (Figure 1d) in the PEM–*L. major* amastigote model were determined at three different culture media flow rates, (i) static (0 m/s), (ii) low flow (1.45 × 10^−9^ m/s; cells placed at bottom of well) and (iii) high flow (1.23 × 10^−7^ m/s; cells placed on top of inserts) [42]. EC_50_ and all measurable EC_90_ values were altered by varying amounts as the speed of media perfusion was increased (Table 2). However, no significance was determined between the EC_50_ or EC_90_ values of any of the drugs, except for a significant change in the EC_50_ of miltefosine between the static and low flow conditions (one way ANOVA *p* < 0.05). The difference by which the EC_50_ values increased when higher media flow rates were applied is small relative to the larger differences noted between the EC_90_ values of amphotericin B and sodium stibogluconate. Both miltefosine and paromomycin showed increases in the EC_50_ value with a changed flow rate. The increase in media perfusion rates affected the Hill slopes of the dose response curve impacting more on the EC_90_ values compared with other measurements of compound potency.

### 3.2. Effect of Culture Media Flow on Drug Accumulation by Peritoneal Macrophages

To investigate why media perfusion altered the EC_50_ and EC_90_ values, we measured the accumulation of amphotericin B and miltefosine by infected PEMs. Drug accumulation by the cells exposed to different culture media flow conditions was different as shown in Figure 2. When comparing amphotericin B drug accumulation between low and high flow conditions a statistical difference is observed (*p* < 0.05 one way ANOVA). A time-dependent and media perfusion-dependent effect could be observed for both drugs, accumulation being significantly higher in the static system relative to the media perfusion systems, after 24 h (Table 3 and Table 4).

Accumulation of amphotericin B (AmB) increased over time (Figure 2) (Table 3 and Table 4), with the highest concentration of AmB within PEMs maintained in the static system seen after 24 h. Comparing this value, a reduced accumulation under low culture media flow (*p* < 0.001, two-way ANOVA) can be observed and a further reduction in accumulation under high culture media flow conditions (two-way ANOVA, *p* < 0.0001) is also seen. The intracellular concentrations of miltefosine also increased over time (Figure 2). By 24 h, the static system showed the highest accumulation of drug compared to the low flow media assay (*p* < 0.01, two-way ANOVA), and the high media flow assay (*p* < 0.001, two-way ANOVA). Both data sets show that, the cells maintained under flow conditions have reduced drug accumulation compared to the static culture I.e., higher the flow speed of the media over the surface of the cells, the lower the drug accumulation.

### 3.3. Formation of 3D Structures

Cells plated in the Alvetex scaffold (Figure 3a) settled throughout the scaffold forming 3D connections to the scaffold itself. However, the cells did not form any secondary structure remaining isolated and not in a multicellular complex. In comparison the cells that were plated in the Invitrocue scaffold (Figure 3b) settled into the pores of the scaffold and formed a secondary multicellular structure. Using the descriptions of the different forms that cells take as a secondary structure in 3D cell culture from Kenny et al. [44], we suggest the secondary structure the cells in the Invitrocue scaffold resembles a grape-like structure.

### 3.4. Establishment of 3D Cell Culture Leishmania Infection Model

To investigate the influence of dimensionality on infection rates, experiments were conducted using two strains of fluorescently labelled parasite. The results (Figure 4) show that percentage infection by *L. major* or *L. amazonensis* were similar in both the Invitrocue sponge and the 2D cell culture. In contrast, the Alvetex scaffold showed much lower levels of macrophage infection. *L. major* infection rates in 3D cell cultures showed greater variability compared to those of *L. amazonensis*. For subsequent experiments, it was decided that the Invitrocue sponge would be used in 3D cell culture systems, as its infection rates matched those seen in 2D cell cultures.

### 3.5. Influence of 3D Cell Culture Set-Up on Anti-Leishmanial Drug Activity

Activities of AmB and of miltefosine were determined against the two fluorescent parasite strains, *L. amazonensis* DSRed2 and *L. major* mCherry. The proportion of infected PEMs as well as the number of parasites per macrophage (Figure 5, Figure 6, Figure 7 and Figure 8) at each drug concentration were counted and used to calculate activity. The EC_50_ values for the experiments using *L. major* and *L. amazonensis* are summarised in Table 5 and Table 6 respectively. Volocity refers to automated counting of images utilising the software package of the same name. Computer counting refers to automated counting of images using the ImarisCell module of Imaris 8.2.0.

Dose response relationships were determined by both methods and both showed equivalent reduction of infections in the 3D and 2D cultures for both *L. major* and *L. amazonensis.* The only significant difference noted with amphotericin B was between the 3D assay counted with the Volocity method and the 2D manual counting (ANOVA one-way *p* < 0.05). When using miltefosine no statistical difference was seen for *L. major* between any of the counting methods (*p* > 0.05). Similarly, when using amphotericin B on *L. amazonensis* infected cells, no significant difference was seen between the values produced by any of the counting methods (*p* > 0.05). A significant difference was seen between the two different methods of automated counting (ImarisCell 8.2.0 referred to as computer counting and Volocity), which had been used to analyse the 3D images taken over the course of the experiments, when comparing the values produced for miltefosine against *L. amazonensis* (ANOVA, *p* < 0.05). 

Similarly, when the number of parasites per peritoneal macrophages is considered, there was no significant difference between the 3D and the 2D cell culture systems, and similar reductions in parasites per peritoneal macrophages were achieved by the two drugs for both *L. major* and *L. amazonensis* (two-way ANOVA, *p* < 0.05).

#### 3.5.1. Evaluation of the Infection Potential in iPSC Macrophages 

When investigating the potential of iPSC derived macrophages as a host cell for *Leishmania* parasites, we initially determined *Leishmania* infection rates in the iPSC macrophages in comparison to other types of macrophage that are commonly used (THP1, HBMM and MBMM) to ensure there were comparable levels of infection established. Four *Leishmania* species, at a range of parasite to host cell ratios, were used in this study to determine if % host cell infection of iPSC derived macrophages were equal utilising different *Leishmania* species (Figure 9).

THP1 cells consistently showed the lowest *Leishmania* percentage infection rates (Figure 9). In contrast, iPSC derived macrophages showed the highest infection rates, especially at the lower parasite: host cell ratios, for both *L. major* JISH and *L. amazonensis*. MBMM and HBMM showed similar infection rates with *L. major* JISH and *L. amazonensis*. However, MBMM showed higher infection rates for *L. mexicana* compared to HBMM and the opposite was seen for *L. major* mCherry, where HBMM showed higher infection rates compared to MBMM. Overall, we show that all four species of *Leishmania* established high levels of infection in iPSC macrophages.

#### 3.5.2. Evaluation of Anti-Leishmanial Drug Activity in iPSC Derived Macrophages 

Initially we established the parasite: host cell ratio that caused a percentage infection of more than 80% after 72 h in the untreated controls (Table 7). This ratio differed for each parasite strain and cell type and the ratio shown in the table was subsequently used to infect the different cell types (iPSC derived macrophages, HBMM and MBMM and THP1 cells). After 24 h, the infection was measured to check that the infection was established (Table 8). Then the infected host cells were exposed to a range of concentrations of amphotericin B, miltefosine and sodium stibogluconate (Sb^V^) for 72 h. Dose responses were determined (see Appendix A) and from these, EC_50_ and EC_90_ values were determined (Table 9, Table 10 and Table 11).

The EC_50_ values for amphotericin B were similar in all cell types (Table 9, Table 10 and Table 11). EC_50_ values for miltefosine were similar for *L. major* JISH or *L. mexicana* infections. However, for *L. major* mCherry responses are more variable and miltefosine proved less effective against *L. amazonensis*. EC_50_ values for sodium stibogluconate were similar between cell types with a couple of outliers (*p* < 0.01). In MBMM high EC_50_ values were determined for sodium stibogluconate on *L. major* JISH or *L. major* mCherry infections. THP1 cells infected with *L. mexicana* parasites and treated with sodium stibogluconate also gave a higher than average EC_50_. iPSC derived macrophage values were mostly similar to HBMM.

## 4. Discussion

Over the past decade anti-leishmanial drug discovery has been significantly enhanced by the screening of compound libraries against intracellular *Leishmania* amastigotes mainly in HCS assays, with over 4 million compounds now tested [45], many hits identified and several novel candidates in pre-clinical and clinical development. In vitro assays also have a key role in lead identification, lead optimization and the determination of structure activity relationships (SARs). We have focussed on the development of more complex in vitro assays, based upon a standard amastigote-macrophage model, to investigate whether any offer suitable alternatives for the more demanding studies to differentiate hits and leads. In other disease areas both microfluidics and tissue engineering have been used to adapt assays for potency and PK-PD studies, for example 3D models in cancer cell drug sensitivity [46,47] and in cytotoxicity assays [48]. Microfluidics and flow have been used to analyse PK/PD relationships of antibacterial [49] and antimalarial [26] compounds. Recently, both 3D and media perfusion were used to model drug combinations against *M. tuberculosis* in a hollow fibre bioreactor model [50]. None of these methodologies have so far been used to explore the drug activities of anti-leishmanial compounds, which still utilise models and approaches that were established over 30 years ago [51]. The primary functions of in vitro models are to identify compounds with potency and selectivity, to provide structure activity relationship data and to ensure that promising compounds progress to in vivo studies.

### 4.1. Influence of Culture Media Flow on Drug Activity

All four standard anti-leishmanial compounds (amphotericin B, miltefosine, paromomycin sulphate and sodium stibogluconate) showed varying levels of activity in static, slow culture medium perfusion and faster perfusion assays, but with a pattern of greater potency in static assays compared to media flow. This pattern is not obvious when comparing EC_50_ values which are similar. However, the shift is larger but still not significant (*p* > 0.05) when comparing EC_90_ values which increase with the rate of media perfusion. This reflects changes in the Hill slope, in that the faster the flow the less steep the Hill slope is, suggesting that the higher the flow the less competition for receptor binding is present, reducing the Hill co-efficient closer to 1 which is a sign of 1:1 binding without any other intervention. A higher Hill slope is often a sign of multiple molecules binding to the same target, in this case the compound to the parasite. For anti-infectives, the EC_90_ values of compounds are more important as the ultimate aim is to eliminate the infection from tissues. Of the several possible explanations for the reduced activity, from cell metabolic rates to oxygen tension [52] we investigated only the change in drug accumulation by macrophages under the different media perfusion rates. This demonstrated a significantly lower accumulation of the two drugs studied, amphotericin B and miltefosine, by infected PEMs at high fluid speeds. We did not determine whether this was driven mainly by the host cell or intracellular parasites. Media perfusion could affect the binding of the drug molecules to their receptors through either a reduction in contact time over the surface area or by lowering the probability of molecules coming in contact with the surface receptors that lead to uptake into the cell by endocytosis. Both amphotericin B and miltefosine are highly protein bound [53] and therefore endocytosis mechanisms are also important. Previously [42] we have shown that the media perfusion system created zones of re-circulation that could lead to a localized lower drug concentration. 

### 4.2. Influence of 3D on Cell Culture Infection and Drug Activity

Initially, we sought to establish which of two scaffolds selected was capable of establishing an infection suitable for drug studies. Based upon percentage macrophage infection after 72 h, all three systems (2D, 3D Alvetex and 3D Invitrocue) provided for increasing infection levels, although variability was higher in the 3D scaffolds compared to the 2D system. The Invitrocue scaffold showed both similar infection rates to the 2D assay, in contrast to the Alvetex scaffold which showed significantly lower infection rates, and impaired formation of a multicell complex. This could be due to the smaller pore size in the Alvetex scaffold (34–40 vs. 80–150 µm in Invitrocue) resulting in an increase in contact in the latter between the scaffold and the peritoneal macrophage surface. The receptors responsible for cell entry are arguably presented and localised better in the Invitrocue scaffold compared to the Alvetex one. For this reason, subsequent experiments were conducted using only the Invitocue CelluSponge scaffold.

Although the only statistically significant differences in drug potencies between the 2D and 3D in vitro infection models was seen when treating *L. major* infected macrophages with amphotericin B, all of the EC_50_ values determined in the 2D and 3D assays are within the range of values reported in the literature [54,55]. 

We used two methods to determine infection levels and drug activity in the 3D cultures: (i) Invitrocue algorithm, and (ii) counting via Volocity software. Discerning the exact number of parasites per peritoneal macrophage in a highly infected cell was difficult by normal light microscopy. Using two algorithms we found that the Volocity algorithm was far more user driven and allowed for user intervention to adjust the parameters and more accurately count the objects within the image. For both types of analysis, the fluorescent signal from the individual parasites merged meaning it was sometimes impossible to segment individual amastigotes. When the average parasite burden per host cell was compared, it was evident that amphotericin B was more effective in clearing amastigotes in contrast to miltefosine which was able to reduce the percentage of infected cells but with infected cells showing a higher parasite burden. Future studies should use imaging equipment with higher level of sensitivity.

Our results are similar to those seen in studies on other pathogens when the activity of compounds was compared between 2D and 3D cell cultures. Bielecka et al. [56] tested anti-tuberculosis drugs in 2D and 3D systems and showed that whilst most compounds showed similar activities in the 3D and 2D cultures, there were a small number of compounds that showed different activities in 3D compared to 2D assays. Cancer studies have also shown differences in sensitivity to drug exposure between cells grown in 2D and 3D, for example, A431.H9 cells grown in 2D and 3D show differences in viability when treated with the same concentrations of 5-fluorouracil and tirapazamine [57]. Another study [58] showed that Her2+ breast cancer cell lines grown in 3D exhibited a lower susceptibility to doxorubicin compared to 2D controls. Cells grown in 3D and then treated with tamoxifen were less susceptible to the cytotoxic effects of the drug than cells grown in 2D. This research highlights the fact that cells in 3D do not necessarily exhibit a higher drug susceptibility, but rather it is a combination of the specific drug and the cellular environment that influences the experimental outcome.

### 4.3. The Potential for Use of iPSC Macrophages in Anti-Leishmanial Drug Assays

It has previously been shown that the type of macrophage used can significantly impact on drug activity against intracellular *Leishmania* amastigotes [59,60]. Initially we established that all four cell types we were investigating could maintain adequate levels of infection with *L. major* JISH, *L. major* mCherry, *L. amazonensis* and *L. mexicana* species/strains. This is the first report that iPSC derived macrophages have been shown to host a viable and dividing infection by a variety of *Leishmania* species. It has been previously shown that iPSC derived macrophages and THP1 cells are both able to phagocytose inert particles at similar rates [41]. One study [61] investigated differential gene expression upon polarisation of all the four macrophages types used in our studies concluding that the iPSC derived macrophages were most similar to the HBMM. The results of our experiments are similar demonstrating that infection rates for IPSCs were most similar to HBMM. Spiller et al. [61] also showed that the MBMM were the most different to the other cell types. This latter result was not replicated in our studies where THP1 cells have the largest difference in response to *Leishmania* parasites. Further confirmatory studies are required with normalized initial infections to overcome the inoculum effect. 

Previous studies [59,60] using THP1 cells, human peripheral blood mononuclear cell derived macrophages and mouse PEMs infected with *L. donovani*, showed that host cell type could have a significant impact on the efficacy of standard anti-leishmanial drugs. We have now added iPSC cells to this list of macrophage types showing that these respond to anti-leishmanial drugs in a similar fashion to that observed in other cell types commonly used in in vitro assays [59]. Similar dose-response curves and patterns in reduction of infection are seen in all cell types, when using the same parasite species. The EC_50_ values obtained are similar to values reported in the literature [54,60,62]. 

Although iPSC derived macrophages are more difficult and expensive to obtain than cell lines, they can be utilised in large scale assays whilst reducing the heterogeneity of macrophages obtained from commercial sources from different donors. Compared to primary cells, their expansion potential is a great advantage, but if not maintained correctly they can occasionally mutate into a cancer phenotype due to their unrestricted growth cycle [63], caused by the addition of stem cell transcription factors. The scientific literature on iPSCs, and more specifically iPSC derived macrophages, is relatively small at present and none have made similar inter-macrophage comparisons. Our paper focuses on iPSC derived macrophages as a pathogen infection model specifically showing that they can be infected by *Leishmania* and respond to anti-leishmanial compounds as expected. 

## 5. Conclusions

The *Leishmania* amastigote–macrophage model has been central to drug discovery assays for nearly 40 years, with modifications over the past decade to enable HCS which has driven the identification of several novel anti-leishmanial compounds. Our approach to investigate the adaptability and predictivity of the model to derive further important data other than compound potency has previously led to data on the use of different macrophage populations [60], and rate of division of amastigotes [13], and here adds to three other refinements—(i) culture media flow, (ii) 3D cell culture, and (iii) iPSC macrophages. Although these refinements do not support the need for any radical change to routine assays, the media flow system causes subtle differences in anti-leishmanial activities and offers opportunities for in vitro PK/PD studies while the research around iPSC macrophages with the opportunity for genetic and transcriptional analysis and modulation is only just beginning. The initial aim of our studies was to determine whether these additional layers of complexity to in vitro assays would enable improved prediction of in vivo activities in rodent models of infection. This was not achieved as some models either proved to be complex and time consuming [42] or did not show significant difference to established assays. 

## Figures and Tables

**Figure 1 microorganisms-08-00831-f001:**
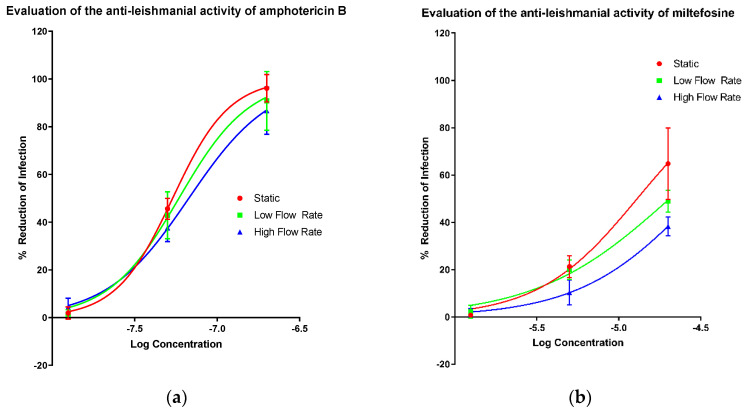
Dose-response curves showing the reduction in infection of mouse peritoneal macrophages (PEMs) caused by amphotericin B (**a**), miltefosine (**b**), paromomycin (**c**) and sodium stibogluconate (**d**) with a variable slope and minimum and maximum set to 0 and 100% respectively. Infection was measured by manual counting using a microscope and the drug-induced percentage reduction of infection was calculated in comparison to untreated controls. Three different experimental conditions resulting in three different media perfusion speeds were used: static (0 m/s), low flow (1.45 × 10^−9^ m/s) and high flow (1.23 × 10^−7^ m/s). *n* = 9. Error bars show SD.

**Figure 2 microorganisms-08-00831-f002:**
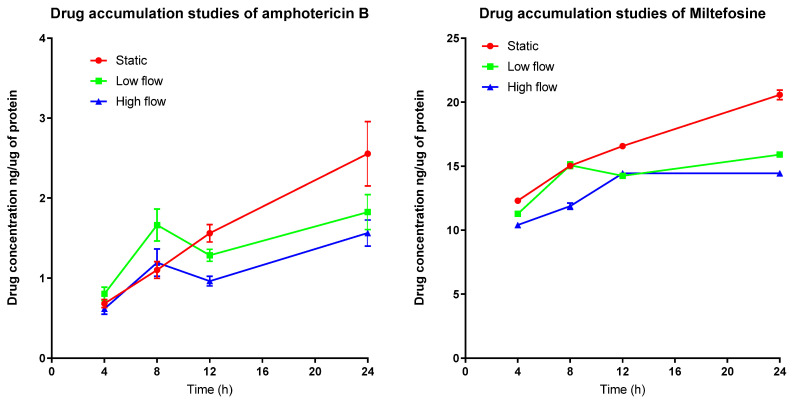
Accumulation of amphotericin B (**left**) and miltefosine (**right**) over time in PEMs maintained either in static condition or in the QV900 media perfusion system Three different experimental conditions resulting in three different media perfusion speeds were used: static (0 m/s), low flow (1.45 × 10^−9^ m/s) and high flow (1.23 × 10^−7^ m/s). *n* = 9, Error bars show SD.

**Figure 3 microorganisms-08-00831-f003:**
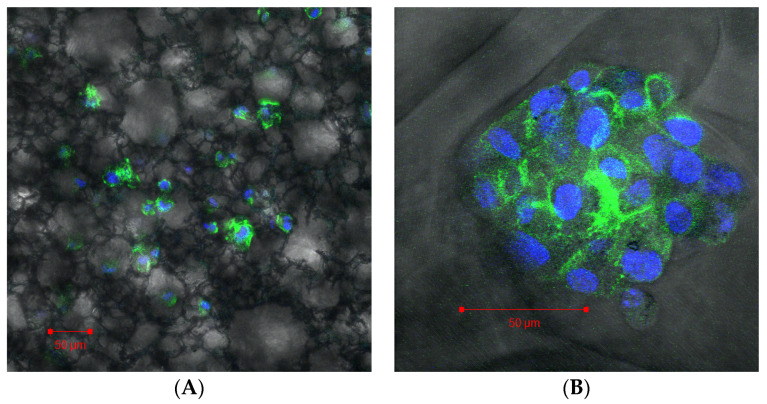
Confocal image of THP-1 cells grown in an Alvetex scaffold (**A**) and Invitrocue scaffold (**B**). Cells stained with DAPI (Blue) and phalloidin (Green).

**Figure 4 microorganisms-08-00831-f004:**
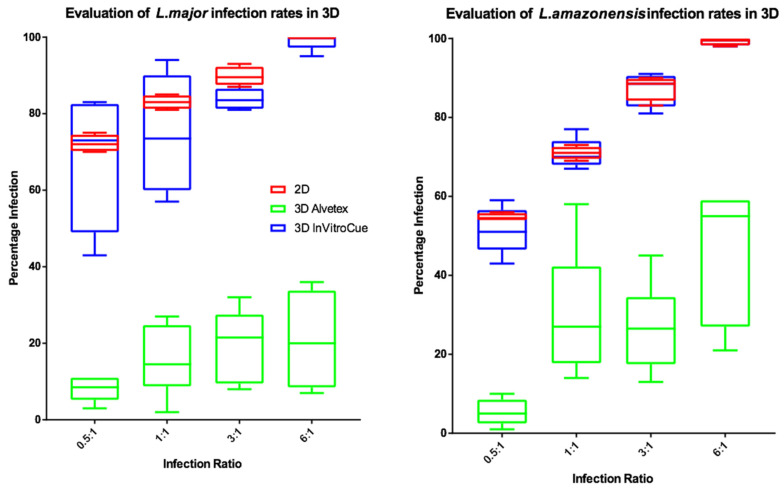
Box and whisker plots showing the infection rates of *L. major* (**left**) and *L. amazonensis* (**right**) in 2D (*n* = 6) or 3D (*n* = 6) cell cultures, using Invitrocue or Alvetex scaffolds. Infection ratio = Parasites: Cells, Red = 2D, Blue = 3D Invitrocue scaffold and Green = 3D Alvetex scaffold. Error bars show minimum and maximum values.

**Figure 5 microorganisms-08-00831-f005:**
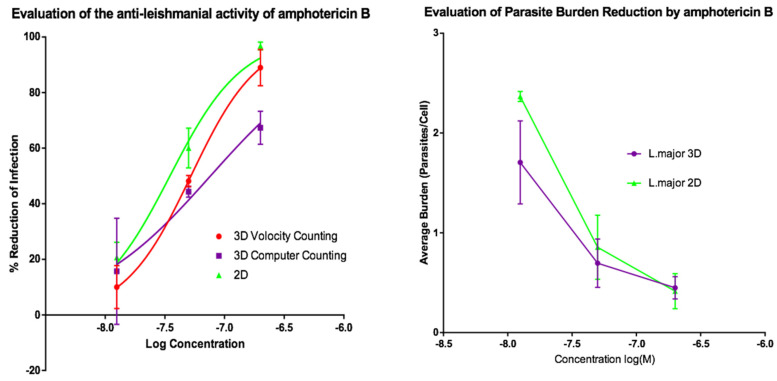
(**Left**) Dose-response curves for 2D (*n* = 6) and 3D (*n* = 6) cell cultures showing the reduction in percentage infection of PEMs by *L. major* produced by treatment using amphotericin B. The percentage reduction in infection calculated based on total infection seen in the untreated controls in either condition. (**Right**) Reduction in *L. major* parasite burden of PEMs for 2D (*n* = 6) and 3D (*n* = 6) cell cultures, produced by dosing with amphotericin B. Error bars show standard deviation.

**Figure 6 microorganisms-08-00831-f006:**
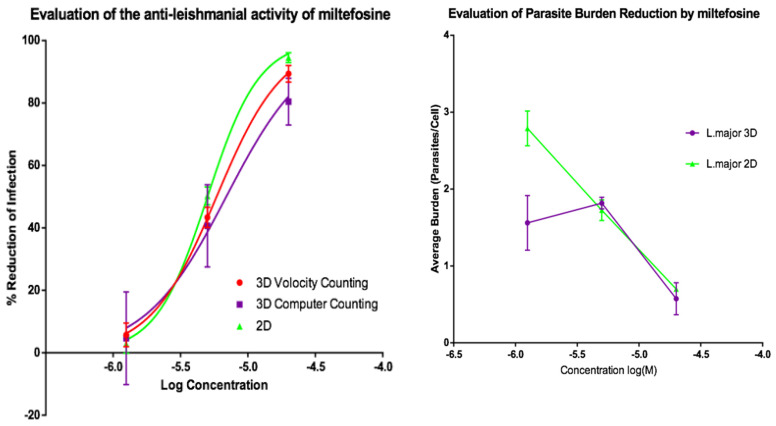
(**Left**) Dose-response curves for 2D (*n* = 6) and 3D (*n* = 6) cell cultures showing the reduction in percentage infection of PEMs by *L. major* produced by treatment using miltefosine. The percentage reduction in infection calculated based on total infection seen in the untreated controls in either condition. (**Right**) Reduction in *L. major* parasite burden of PEMs for 2D (*n* = 6) and 3D (*n* = 6) cell culture, produced by dosing with miltefosine. Error bars show SD.

**Figure 7 microorganisms-08-00831-f007:**
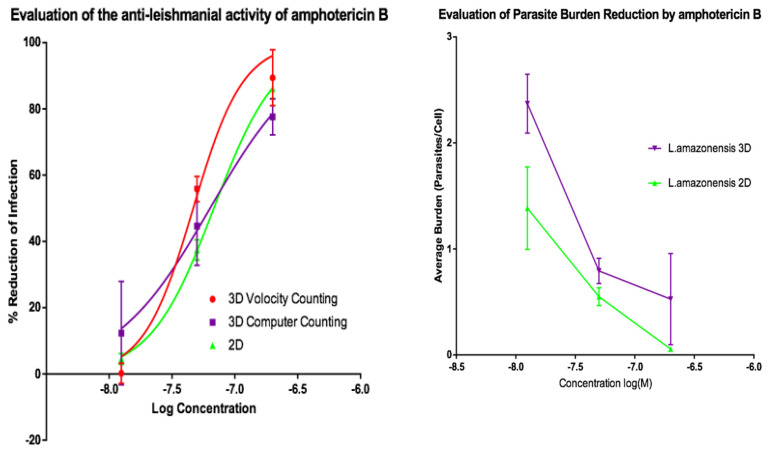
(**Left**) Dose-response curves for 2D (*n* = 6) and 3D (*n* = 6) cell culture showing the reduction in percentage infection of PEMs with *L. amazonensis* produced by treatment using amphotericin B. The percentage reduction in infection calculated based on total infection seen in the untreated controls in either condition. (**Right**) Reduction in *L. amazonensis* parasite burden of PEMs for 2D (*n* = 6) and 3D (*n* = 6) cell culture, produced by exposure to amphotericin B. Error bars show standard deviation.

**Figure 8 microorganisms-08-00831-f008:**
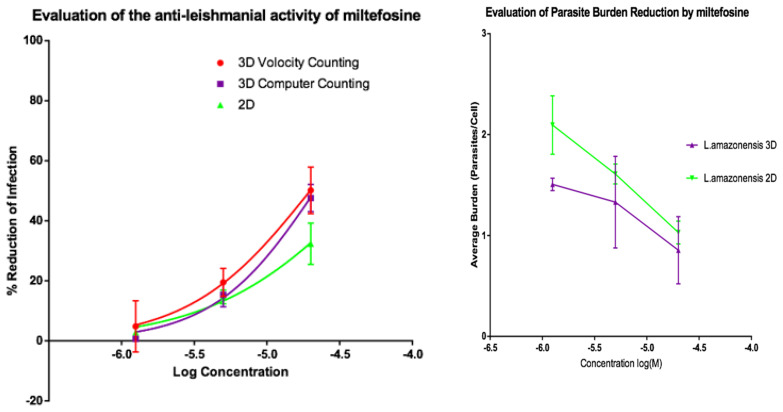
(**Left**) Dose-response curves for 2D (*n* = 6) and 3D (*n* = 6) cell culture showing the reduction in percentage infection of PEMs with *L. amazonensis* produced by dosing with miltefosine. The percentage reduction in infection calculated based on total infection seen in the untreated controls in either condition. (**Right**) Reduction in *L. amazonensis* parasite burden of PEMs for 2D (*n* = 6) and 3D (*n* = 6) cell culture, produced by dosing with miltefosine. Error bars show standard deviation.

**Figure 9 microorganisms-08-00831-f009:**
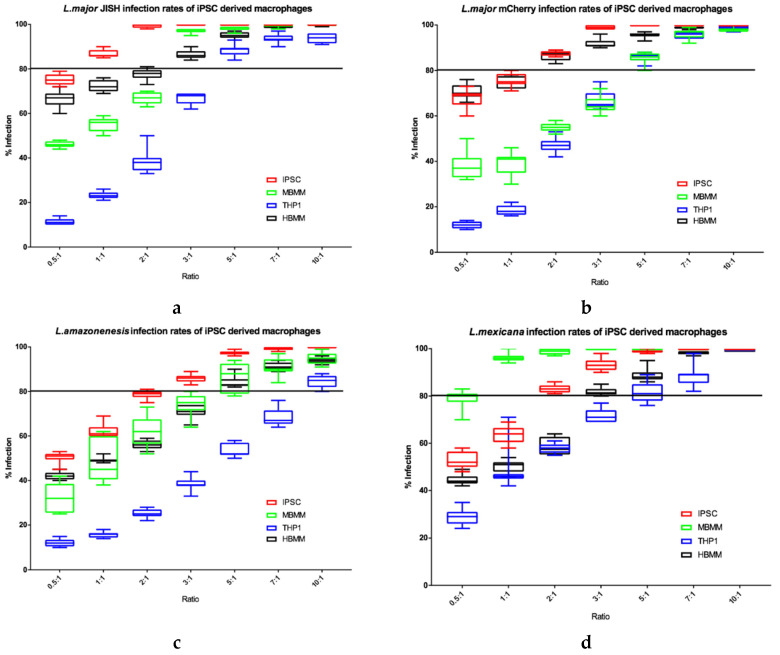
Percentage infections of *L. major* JISH (**a**), *L. major* mCherry (**b**), *L. amazonensis* (**c**) and *L. mexicana* (**d**) after 72 h incubation in each of the four cell types. *n* = 9. Line is drawn at 80% infection. Error bars show minimum and maximum values.

**Table 1 microorganisms-08-00831-t001:** Four standard drugs used in the assays.

Compound	Supplier	Catalog Number
miltefosine	Nycomed, Aldwych, UK	Custom
amphotericin B	VWR international, Lutterworth, UK	E437
paromomycin sulphate	Sigma, Gillingham, UK	P9297
sodium stibogluconate	Sigma, Gillingham, UK	S5319

**Table 2 microorganisms-08-00831-t002:** EC_50_ and EC_90_ values of the four drugs under three different conditions of culture media flow. Mean and 95% confidence intervals are shown. NC = Not calculated.

Condition	Static		Low Flow		High Flow	
	EC_50_	EC_90_	EC_50_	EC_90_	EC_50_	EC_90_
Amphotericin B (nM)	54 (51–57)	124 (100–150)	63 (52–67)	196 (114–244)	70 (61–75)	273 (185–305)
Miltefosine (μM)	12 (11–15)	NC	21 (18–23)	NC	30 (26–34)	NC
Paromomycin (μM)	85 (64–111)	NC	198 (162–254)	NC	188 (158–230)	NC
Sb^V^ (μg/mL)	223 (220–237)	368 (350–465)	228 (214–242)	532 (436–535)	212 (193–233)	709 (572–892)

**Table 3 microorganisms-08-00831-t003:** Concentrations of amphotericin B in infected PEMs (ng/µg of protein) maintained at the three different speeds of media perfusion. Means and 95% confidence intervals are shown.

Time (h)	ng of Drug Per µg of Protein in Cell Lysates		
	Static	Low Flow	High Flow
	0 nm/s	1.45 ± 0.01 nm/s	123 ± 0.001 nm/s
4	0.68 (0.57–0.80)	0.80 (0.61–1.00)	0.62 (0.46–0.77)
8	1.10 (0.86–1.34)	1.66 (1.20–2.13)	1.19 (0.80–1.59)
12	1.56 (1.31–1.81)	1.28 (1.11–1.46)	0.96 (0.82–1.10)
24	2.56 (1.63–3.48)	1.83 (1.32–2.33)	1.56 (1.19–1.94)

**Table 4 microorganisms-08-00831-t004:** Concentrations of miltefosine in infected PEMs (ng/µg of protein) maintained at the three different speeds of media perfusion. Means and 95% confidence intervals are shown.

Time (h)	ng of Drug Per µg of Protein in Cell Lysates		
	Static	Low Flow	High Flow
	0nm/s	1.45 ± 0.01 nm/s	123 ± 0.001 nm/s
4	12.3 (12.5–12.1)	11.3 (10.9–11.6)	10.4 (9.9–10.9)
8	15.0 (14.7–15.3)	15.1 (14.4–15.7)	11.9 (11.2–12.5)
12	16.6 (16.4–16.7)	14.2 (14.1–14.4)	14.4 (14.1–14.9)
24	20.6 (19.6–21.5)	15.9 (15.9–15.9)	14.4 (13.9–15.0)

**Table 5 microorganisms-08-00831-t005:** EC_50_ values of amphotericin B and miltefosine against *L. major* mCherry amastigotes in PEMs, in 2D or 3D cell culture. Means and 95% confidence intervals are shown.

	EC_50_		
	3D		2D
	Volocity Counting	Computer Counting	Manual
Amphotericin B (nM)	52.3 (46.6–58.7)	76.2 (37.7–203.5)	34.9 (31.4–38.6)
Miltefosine (μM)	5.85 (5.52–6.21)	6.90 (4.09–12.22)	5.02 (4.88–5.16)

**Table 6 microorganisms-08-00831-t006:** EC_50_ values of amphotericin B and miltefosine against *L. amazonensis* amastigotes in PEMs, in 2D or 3D cell culture. Means and 95% confidence intervals are shown.

	EC_50_		
	3D		2D
	Volocity Counting	Computer Counting	Manual
Amphotericin B (nM)	46.7 (41.5–52.3)	63.5 (36.1–117.0)	68.0 (65.3–70.8)
Miltefosine (μM)	19.8 (16.2–25.9)	21.4 (17.4–28.7)	47.6 (37.2–66.8)

**Table 7 microorganisms-08-00831-t007:** Optimal initial infection ratio for each parasite strain for each of the four different cell types used.

	THP1	MBMM	HBMM	iPSC
*L. major* JISH	5:1	3:1	3:1	1:1
*L. major* mCherry	5:1	5:1	2:1	2:1
*L. amazonensis* dsRed	10:1	7:1	7:1	3:1
*L. mexicana*	7:1	1:1	5:1	2:1

**Table 8 microorganisms-08-00831-t008:** The initial infection after 24 h for each parasite strain for each of the four different cell types used.

	% Infection Levels after 24 h			
	THP1	MBMM	HBMM	iPSC
*L. major* JISH	30 ± 1	55 ± 2	37 ± 6	27 ± 2
*L. major* mCherry	27 ± 4	59 ± 4	32 ± 2	42 ± 1
*L. amazonensis* dsRed	41 ± 4	60 ± 1	70 ± 1	52 ± 1
*L. mexicana*	45 ± 4	26 ± 2	68 ± 7	69 ± 1

**Table 9 microorganisms-08-00831-t009:** EC_50_ and EC_90_ values of amphotericin B (nM) against the different parasites in different host cell types. Means and 95% confidence intervals are shown.

		iPSC	THP1	MBMM	HBMM
*L. major* JISH	EC_50_ nM	35.9 (33.4–38.5)	46.4 (45.8–47.2)	33.5 (31.5–36.5)	33.6 (30.5–NC)
	EC_90_ nM	105.2 (94.2–117.5)	70.5 (66.1–74.2)	62.9 (59.8–65.9)	54.4 (NC–57.0)
*L. major* mCherry	EC_50_ nM	70.7 (68.4–73.2)	69.0 (67.7–69.7)	44.1 (42.2–45.7)	38.2 (36.3–40.1)
	EC_90_ nM	182.0 (165.6–202.0)	102.5 (100.0–105.0)	85.0 (77.3–93.0)	78.5 (72.9–84.1)
*L. amazonensis* dsRed	EC_50_ nM	79.8 (77.6–82.0)	63.4 (61.0–64.6)	60.9 (57.5–64.3)	51.1 (46.9–55.4)
	EC_90_ nM	229.2 (213.3–247.0)	159.3 (148.5–171.7)	251.6 (222.3–286.6)	188.8 (159.7–226.0)
*L. mexicana*	EC_50_ nM	59.8 (58.2–61.3)	76.5 (75.1–78.1)	55.4 (54.0–56.9)	67.8 (65.3–70.3)
	EC_90_ nM	102.4 (95.9–109.4)	144.6 (139.0–150.9)	109.2 (102.0–117.2)	131.5 (121.4–143.6)

**Table 10 microorganisms-08-00831-t010:** EC_50_ and EC_90_ values of miltefosine (µM) against the different parasites in different host cell types. Means and 95% confidence intervals are shown.

		iPSC	THP1	MBMM	HBMM
*L. major* JISH	EC_50_ µM	5.8 (5.2–6.5)	7.7 (7.5–8.1)	6.4 (6.1–6.7)	7.4 (6.9–8.0)
	EC_90_ µM	NC	17.7 (16.5–19.2)	19.2 (17.4–21.4)	34.2 (28.5–41.4)
*L. major* mCherry	EC_50_ µM	15.3 (14.2–16.4)	9.4 (8.5–10.3)	14.9 (12.2–16.9)	4.9 (4.1–5.7)
	EC_90_ µM	NC	NC	NC	16.1 (11.3–23.9)
*L. amazonensis* dsRed	EC_50_ µM	76.6 (64.2–94.9)	28.2 (26.6–30.0)	25.3 (23.5–27.4)	14.3 (13.3–15.4)
	EC_90_ µM	NC	NC	NC	62.2 (52.2–75.6)
*L. mexicana*	EC_50_ µM	5.3 (5.0–5.5)	4.9 (4.6–5.1)	5.0 (4.7–5.3)	6.0 (5.3–6.8)
	EC_90_ µM	17.2 (15.6–18.9)	15.6 (13.9–17.6)	NC	25.1 (19.7–33.0)

**Table 11 microorganisms-08-00831-t011:** EC_50_ and EC_90_ values of sodium stibogluconate (ugSb^V^/mL) against the different parasites in different host cell types. Means and 95% confidence intervals are shown.

		iPSC	THP1	MBMM	HBMM
*L. major* JISH	EC_50_ μgSb^V^/mL	355 (308–415)	186 (178–193)	2770 (2017–4146)	348 (324–375)
	EC_90_ μgSb^V^/mL	NC	NC	NC	NC
*L. major* mCherry	EC_50_ μgSb^V^/mL	220 (209–232)	253 (234–274)	730 (686–787)	434 (397–477)
	EC_90_ μgSb^V^/mL	NC	NC	NC	NC
*L. amazonensis* dsRed	EC_50_ μgSb^V^/mL	306 (266–352)	459 (442–477)	503 (485–524)	255 (238–273)
	EC_90_ μgSb^V^/mL	NC	NC	NC	NC
*L. mexicana*	EC_50_ μgSb^V^/mL	347 (328–366)	1840 (1481–2411)	435 (406–465)	299 (268–332)
	EC_90_ μgSb^V^/mL	NC	NC	NC	NC

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
