# Peer review of "Novel 2D and 3D Assays to Determine the Activity of Anti-Leishmanial Drugs"

_microorganisms, 2020, doi:10.3390/microorganisms8060831_

Round 1
Reviewer 1 Report
The manuscript by O’Keeffe et al. puts forward a very insightful comparative analysis on how different in vitro models of Leishmania-infected macrophages perform in the context of anti-Leishmania drug discovery. This manuscript will be considered suitable for publication after appropriate changes are made.
General notes
- The “Results” section requires grammar and style check.
- Authors present quantitative data either as “average +/- SD”, “average (minimum - maximal)” or “mean (lower - higher 95% confidence values)”. For consistency, and to facilitate inter-experiments comparative analyses, authors should adopt only one format throughout the manuscript.
- Flow speed is given either as ul/min, nm/s or um/s. Please, normalize this: e.g. nm/s throughout the entire document (including “Materials and Methods”).
Major comments
- Lines 174-188. According to the experimental design described here, infection with Leishmania is initiated already in the presence of drugs and in the presence of active flow. This raises some questions: i) This model differs from all others tested in this manuscript, in which drugs are added to macrophages AFTER infection with Leishmania. By adding drugs PRIOR to infection can authors exclude their effect against promastigotes? ii) Is the capacity of Leishmania to invade macrophages affected by the flow rate? Have authors addressed this?
- Lines 270-271. “low flow ([…] cells placed at bottom of well) and (iii) high flow ([…] cells placed on top of inserts)”. These are important experimental details, which are nonetheless missing in the "Materials and Methods". Please provide more information in this section, making reference to the flow rates that were tested.
- Lines 277-278. “The difference by which the EC50 values increased when higher media flow rates were applied is small relative to the larger differences noted between the EC90 values of each drug”. This is indeed the case of amphotericin B and sodium stibogluconate, but not of miltefosine, paramomycin. For the latter, the EC50 is affected by the flow rate – it increases 2x for miltefosine and 9x for paramomycin as the conditions go from static to high flow. Please reconsider and rewrite accordingly.
- Line 281. “maximum and minimum set to 0 and 100% respectively”. Two questions: i) Do authors mean “MINIMUM and MAXIMUM set to 0 and 100% respectively”? i) In the graphs, not all values vary between 0 and 100%. Were the minimum and maximum really set between 0 and 100%?
- Tables 3 and 4 are not cited in the manuscript. Either remove the tables or include a text referring to them.
- Table 4. The values in this table - in the range of 0.19-0.35 ng miltefosine/ug proteins - do not coincide with the ones plotted in the corresponding graph (Fig.2, right panel) - in the range of 10-20 ng/ug. This is not merely an issue of magnitude: the numerical values are different even if we try to correct them for the same magnitude (e.g. accumulation of miltefosine at 4 hrs under high flow: 0.19 in the table vs. ~10 in the graph). Please provide an explanation to this and present consistent data.
- Figure 3. Remove panel C, as it does not refer to original results from this manuscript. Remove reference to Fig. 3C from the text as well.
- Lines 341-342. “Volocity refers to automated counting of images utilising the software package of the same name.” In this sentence, authors should also include an explanation about the meaning of “computer counting” (Fig. 5-8). I assume that it refers to the ImarisCell module of Imaris 8.2.0, but readers must go all the way back to the “Materials and Methods” section to get this info.
- Tables 5 and 6. The EC50 and EC90 values for amphotericin should be provided in nM (for consistency with the other manuscript sections).
- Line 409. It is not clear why authors refer to Table 8 in the sentence “from these, EC50 and EC90 values were determined (Tables 8-11).” Table 8 shows the initial infection after 24 hours, not EC50/90 values.
- Table 8 presents very important data and authors must include a proper text and discussion around it. Don’t authors expect the initial infection level to influence final EC50 readout? In other words: isn’t the potency of a given drug expected to be higher when initial infection rates are lower?
- Results in Table 9-11 are extremely dense and difficult to interpret. I suggest authors to plot them in graphs. Moreover, Tables 9-10 do not indicate the units of EC50 and EC90 values.
- Line 423. “The EC50 values for amphotericin B were similar in all cell types (Tables 8 -11)”. Please reconsider your data and rewrite the sentence. E.g. L. major mCherry iPCS vs. hBMM: 70.7 vs. 38.2.
- Line 425. “miltefosine proved ineffective against L. amazonensis”. Please rephrase. Table 10 shows that the EC50 value for miltefosine against L. amazonensis is indeed high (76.6) when iPSC are used as host cells. However, with hBMM this parameter drops to 14.3 (not so different for the EC50 values of miltefosine against e.g. L. major Cherry).
- Some EC50/90 values are presented as “mean (lower - higher 95% confidence values)”, e.g. 35.9 (33.4 – 38.5). In some cases, authors indicate that one of 95% confidence values was not calculated (NC), e.g. 33.6 (30.5 –NC). I reckon this is only conceivable if values do not follow a normal distribution. Please, clarify this. I also suggest authors to introduce a brief description of their statistical analyses in the “Materials and Methods” section.
Minor comments
- Lines 234-235. “(using the ImarisCell module of Imaris 8.2.0 (Bitplane A)”. A parenthesis is missing after “(Bitplane A)”.
- Figures 1b and 4-9 have poor resolution.
- Line 292. “Fig.5”. I reckon authors mean “Fig.2”.
- Lines 337-339. “Activities of AmB and of miltefosine WERE DETERMINED against the two fluorescent parasite strains, L. amazonensis DSRed2 and L. major mCherry WERE DETERMINED by measuring reductions (…)”. Rephrase.
- Line 362. “by dosing with” correct to “by dosing with miltefosine”.
- Line 399. “was seen for L. major mCherry, WHEN…” correct to “was seen for L. major mCherry, WHERE…”.
- Line 407. “exposed to a range of concentrations of THREE OF: amphotericin” correct to “exposed to a range of concentrations of amphotericin”.
- Line 527. In the sentence “anti-leishmanial drugs macrophages”, delete the word “macrophages”.
- Lines 427-428. “In MBMM high EC50 values were determined for sodium stibogluconate on L. major JISH or L. major mCherry infections as well as for THP1 cells were infected with L. mexicana parasites”. Please correct grammar.
Author Response
General Points
ï‚· Results section - grammar and style corrected ï‚· For consistency, and to facilitate inter-experiments comparative analyses, mean and 95% confidence values are now used throughout the manuscript. ï‚· Flow speed have all been normalised to nm/s.
Major Comments:
Lines 174-188: The drugs are always added after the Leishmania infection is established. We thank the reviewer for pointing out this mistake and have corrected.
Lines 270-271: Further details on flow speeds and set up have been added to the M & M section (182-184). The complete explanation and background work into flow speeds and how they were determined is in O'Keeffe et al 2019 which is referenced.
Lines 277-278: This sentence has been edited to reflect that the EC50 value does change when flow is applied.
Line 281: The minimum and maximum was set to 0 and 100% respectively and this has been changed in the text. In the graphs (Figure 1) the minimum and maximum were set to 0 and 100%, and extrapolations made through Prism software.
Tables 3 and 4: cross reference has been added in the text.
Table 4: We thank the reviewer for pointing this mistake. All the data in the tables is now given with to 95% CI intervals and changed to that data matches the graphs and was produced by the same prism data file as the graph images.
Figure 3: We have removed the diagram in panel C from this Figure as suggested by the Reviewer. However, the comparative nature of 3D cell clusters has been determined by others and is commented on in the text with the reference retained.
Lines 341-342: Computer counting refers to the ImarisCell module of Imaris 8.2.0; this has been added to the text.
Tables 5 and 6: The EC50 values for amphotericin in Tables 5 and 6 have been changed to nM for consistency.
Line 409: This has been changed with an added reference to table 8 in lines 406/7 indicating that table 8 is a measure of the infection after 24 hours.
Table 8 “presents very important data and authors must include a proper text and discussion around it. Don’t authors expect the initial infection level to influence final EC50 readout? In other words: isn’t the potency of a given drug expected to be higher when initial infection rates are lower?”
The 24 hour infection levels are shown in Table 8; by 72 hours all cells in control cultures had infection levels of 80+% . The need to resolve the impact of the different 24 hr levels of infection on the inoculum effect are now added to the discussion (lines 540-541).
Graphical results for tables 9-11 are shown in the supplementary material. Tables 9, 10, and 11: Added units of drug concentration to each table and in the table legend.
Line 423: The EC50 values were not statistically different between cell types (Tables 9 -11, p value >0.05) although differences were observed eg . E.g. L. major mCherry iPCS vs. hBMM: 70.7 vs. 38.2.
Line 425: Sentence has been addressed and re-phrased.
Tables/Figures: “In some cases, authors indicate that one of 95% confidence values were not calculated (NC), e.g. 33.6 (30.5 –NC)”. The abbreviation NC was used for cases where the statistical analysis conducted by the Prism programme returned no values for the 95% confidence limits, probably due to the data not following a normal distribution.
Statistical analysis: a brief description of the statistical analysis and programmes used has been included in the materials and methods section.
Minor Comments:
Figures 1b and 4-9: Images have been exported from Graphpad prism as TIF files at 1200 dpi which is maximum resolution. They have been submitted as separate files for the final version so will hav improved resolution.
Lines 234-235: A parenthesis has been added after “(Bitplane A)”.
Line 292: Reference to figure 5 changed to reference to figure 2.
Lines 337-339: Sentence rephrased to avoid repetition.
Line 362: Sentence changed as suggested.
Line 399: Sentence changed as suggested.
Line 407: Sentence changed as suggested.
Lines 427-428: Sentence split in two and corrected to make more sense.
Line 527: As suggested the word macrophage has been deleted.

Reviewer 2 Report
The article by O’Keeffe et al. describes alternative assays for evaluating drugs against intracellular leishmania amastigotes. The assays are interesting but, as indicated by the authors, none of them improves those that are actually in use. There is an strong technical effort for testing alternative procedures, including several sources of macrophages and culture techniques. The main value of this article consists in communicating that these procedures do not increase the overall quality of the methods currently in use.
My major concern with this manuscript is that there is not any “a priori” statement about which criteria should be used to define the quality of the assays. These criteria are needed to make decisions about their quality. For example, concordance of “in vitro” and “in vivo” results, as in vitro assays are designed to obtain similar information than that from in vivo but with much simpler experimental designs.
Minor issues:
Line 31. With an initial infection ratio of 0.5 parasites per host cell, the authors describe infection rates of 75% macrophages. The authors should explain if amastigotes grow in the initially infected macrophages and, after lysis, there is a second round of infection.
Line 130. The origin of the Thp1 cells is not indicated.
Line 290. p values given in the text are not assigned to any specific comparison among samples.
Line 292. Figure 5 in the text should be changed to figure 2.
Line 316. CelluSponge scaffold is used in the main text and the Invitrocue scaffold is used in the figure. The same name should be used for clarity.
Line 337. This sentence needs to be rewritten.
Line 409. Table 8 does not show EC50 or EC90 values.
Tables 9, 10, and 11 do not include the units of drug concentration.
Author Response
Major Comments:
We thank the reviewer for requesting this clarification. A paragraph has been added to the conclusion (558-564) and a sentence (109) to the introduction. The in vitro assays were indeed designed to investigate whether other more complex “in vitro” systems were more predictive of “in vivo” results. In this regard the aims of the research were not achieved as model establishment was time consuming and complex (flow system) or did not add to the assays already established in many laboratories (see conclusion).
Minor Comments:
Line 31: The experiments were limited to 96 hours and there was no microscopic evidence of macrophage burst and re-infection. In all experiments level of infection was determined at 24 hours prior to drugging, and then again 72 hours later in controls; this shows amastigote increase in numbers.
Line 130: The origin of the THP1 cells has been added.
Line 290: p values have been assigned to enable specific comparison between amphotericin B at high and low flow conditions. Further editing of lines 299-302 have been made to avoid repetition and expand the statistic test to a two-way ANOVA.
Line 292: Reference to figure 5 changed to reference to figure 2.
Line 316: References to the Cellusponge have been removed and replaced by either the term Invitorcue scaffold or the Invitrocue Cellusponge.
Line 337: Sentence was split in two and changed to make more sense.
Line 409: Edited so the table reference does not include table 8. Added a reference to table 8 in lines 406/7 indicating that table 8 is a measure of the infection after 24 hours.
Tables 9, 10, and 11: Added units of drug concentration to each table and in the table legend.
